# CommonsenseQA 2.0: Exposing the Limits of AI through Gamification

**Alon Talmor**[1]  **Ori Yoran**[1,2]  **Ronan Le Bras**[1]  **Chandra Bhagavatula**[1]
**Yoav Goldberg**[1]  **Yejin Choi**[1,3]  **Jonathan Berant**[1,2]
[1]The Allen Institute for AI, [2]Tel-Aviv University, [3]University of Washington
{ronanlb,chandrab,yoavg,yejinc,jonathan}@allenai.org
{alontalmor,oriy}@mail.tau.ac.il

## Abstract

Constructing benchmarks that test the abilities of modern natural language understanding models is difficult – pre-trained language models exploit artifacts in benchmarks to achieve human parity, but still fail on adversarial examples and make errors that demonstrate a lack of common sense. In this work, we propose gamification as a framework for data construction. The goal of players in the game is to compose questions that mislead a rival AI, while using specific phrases for extra points. The game environment leads to enhanced user engagement and simultaneously gives the game designer control over the collected data, allowing us to collect high-quality data at scale. Using our method we create CommonsenseQA 2.0, which includes 14,343 yes/no questions, and demonstrate its difficulty for models that are orders-of-magnitude larger than the AI used in the game itself. Our best baseline, the T5-based UNICORN with 11B parameters achieves an accuracy of 70.2%, substantially higher than GPT-3 (52.9%) in a few-shot inference setup. Both score well below human performance which is at 94.1%.

## 1 Introduction

Evaluating models for natural language understanding (NLU) has become a frustrating endeavour. On the one hand, state-of-the-art models achieve human parity on a wide range of NLU benchmarks, which are now generated at unprecedented rate. On the other hand, recurring findings reveal that such models are brittle when provided with out-of-domain [1, 2, 3] or adversarial examples [4, 5, 6, 7, 8, 9, 10], and make silly mistakes that contradict basic notions of human common sense. It is almost as if NLU models learn to solve datasets without solving the underlying tasks. This status quo calls for new innovations in the area of benchmark design.

To generate more challenging datasets, recent research proposed algorithmic methods for bias reduction, such as adversarial filtering [11, 12, 13, 14, 15]. While helpful in mitigating bias and reducing dataset artifacts in data that has already been generated, such approaches are fundamentally post-hoc, that is, they can only remove examples, but do not direct the example generation pipeline to create difficult and diverse examples. Indeed, recent research highlights the importance of diverse and hard instances for robust out-of-domain generalization [16, 17]. Thus, a more fundamental solution should be integrated directly into the data creation process. Unfortunately, the vast majority of existing benchmarks are based on crowdsourcing, which are by and large static and non-interactive, leading to datasets that are plagued with shortcuts and artifacts [18, 19, 20, 4]. At the same time, humans can still reliably compose new examples that demonstrate the failing of current AI systems.

In this paper, we proposes gamification as a framework for data creation that can tap into human intelligence more effectively. Our framework shares the spirit of model-and-human-in-the-loop

35th Conference on Neural Information Processing Systems (NeurIPS 2021) Track on Datasets and Benchmarks.

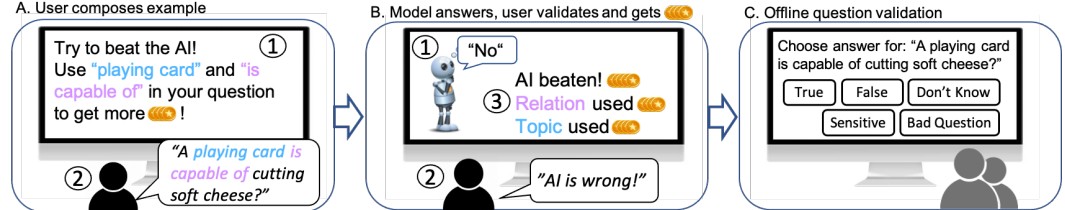

Figure 1: An overview of our approach for data collection through gamification.

approaches, which seek dynamic benchmark creation [21], but provides a game environment that enhances user engagement, which in turn leads to high-quality data and low costs.

Fig. 1 shows a high-level overview of our game, which focuses on collecting yes/no questions or assertions. At a high-level, a player is asked to author a yes/no question, is then shown the answer from the AI, and then marks whether the AI was correct or not. The goal of the player is to earn points, which are used as a flexible vehicle for steering the behaviour of the player. First, points are given for beating the AI, that is, authoring questions where the AI is incorrect. This incentivizes the player to ask difficult questions, conditioned on its understanding of the AI capabilities. Second, the player gets points for using particular phrases in the question. This provides the game designer control to skew the distribution of questions towards topics or other phenomena they are interested in. Last, questions are validated by humans, and points are deducted for questions that do not pass validation. This pushes players to author questions with broad agreement among people. Similar to prior work [21, 22, 23], a model is trained on collected examples, which prevents players from using the same types of questions repeatedly, as the re-trained model quickly adapts to such questions.

Using our method, we collected COMMONSENSEQA 2.0 (CSQA2) containing 14,343 yes/no questions. We extensively evaluate models on CSQA2, experimenting with pre-trained models, fine-tuned models, and reading comprehension (RC) models that utilize web snippets extracted from Google search on top of the question itself.

Despite using RoBERTA-Large [24] as the model in the loop, the resulting dataset proves to be challenging for state-of-the-art models that are orders of magnitude larger. Compared to human performance of $94.1\%$, the best performance the models achieve is $70.2\%$ by UNICORN-11B [25], substantially better than $67.8\%$ by T5-11B [26]. Notably, GPT-3 [1] achieves only $52.9\%$, despite being much larger (but without fine-tuning), even after extensive prompt engineering.

Comprehensive analyses offer new insights into the limitations of current neural models; GPT-3 is relatively weaker in causal reasoning, while fine-tuned UNICORN is weaker in size comparisons. In addition, we find that none of the models are robust against relatively simple linguistic or logical variations of the same question.

To summarize, our contributions are:

1. CSQA2: A challenging and diverse new QA dataset, containing 14,343 examples.
2. A new framework for data creation via gamification alongside a model-in-the-loop approach.
3. An empirical evaluation of state-of-the-art pre-trained language models on CSQA2, showing that humans substantially outperform current models.

Our dataset and code are available at `http://allenai.github.io/csqa2`.

## 2 Data Collection through Gamification

We now describe our game environment for eliciting hard and diverse yes/no questions (or assertions). We first describe the game itself (§2.1), and then detail the dataset construction procedure including measures for quality assurance (§2.2).

### 2.1 The Game

Our premise is that a game environment can create an enjoyable experience for players, while enabling control over the quality and type of data obtained through the game's point system. We now describe

the details of the different parts of the game, which are illustrated in Fig. 1: (a) controlled question generation, (b) model-in-the-loop, and (c) question validation. Full print-screens from the actual game are presented in Fig. 2.

**Controlled question generation**   Examples generated in this work are yes/no questions (or assertions) with their correct answer. We use questions since they are a natural medium for players to test capabilities of the AI, but other formats are also possible.

In this step, players are asked to author questions, and get points if the AI errs on the authored questions. This pushes players to generate difficult questions conditioned on their understanding of the abilities of the AI. To give the game designer control over the topic of the question and the reasoning skills required for answering it, we present the player with two phrases, a *topic prompt* and a *relational prompt*, and points are given for incorporating these phrases in the question. For example, in Fig. 1 the topic prompt is *"playing card"*, and the relational prompt is *"is capable of"*, and indeed the player uses them in *"A playing card is capable of cutting soft cheese"*. The prompts not only provide control, but also spark the creativity of the player, which can author the question around tangible concepts.

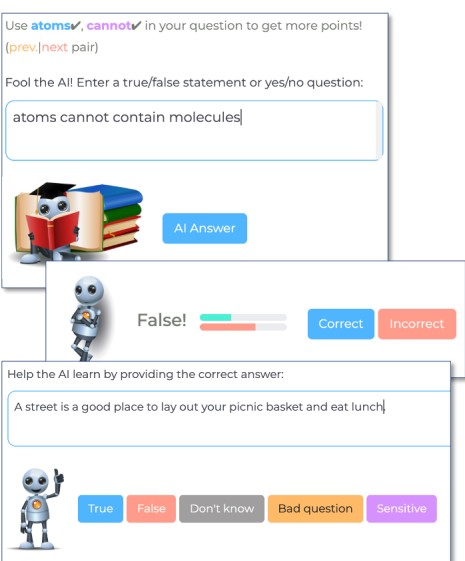

Figure 2: Print-screens of the different game parts.

For topic prompts, we use CONCEPTNET [27], a popular knowledge-graph containing information on commonsense concepts and relations. We choose the 1,875 highest ranking concepts in ConceptNet, and in each question sample one concept uniformly at random. For relational prompts, we use a manually-constructed list of 33 phrases, designed to target a wide set of commonsense reasoning skills, such as understanding meronymy, causality, plausibility, and more. Table 1 shows the full list of relational prompts, their categories and their frequency. While our choice for prompts steers collected data towards commonsense questions, other prompts can be used to get a different distribution.

**Model-in-the-loop**   After a player formulates a question ($A2$ in Fig. 1), e.g., *"A playing card is capable of cutting soft cheese"*, the question is answered by the model ($B1$), and the user gets points if the model is wrong. A potential issue is that players identify a weakness in the current model and only ask questions that target that particular weaknss. Thus, similar to past work [28, 22, 21], we use a model-in-the-loop and re-train the model during data collection. Thus, if players generate many questions exploiting a particular model blind spot, the blind spot will be fixed when the model is re-trained. This leads to more diversity in the collected data.

| Type | RELATIONAL PROMPTS | % |
|---|---|---|
| Taxonomy / other | is, part of, has, have, is a | 20.0 |
| Capable of | is capable of, can, cannot | 14.9 |
| Causality | before, after, because, causes | 11.1 |
| Plausibility | all, some, at least one, at least two, most, none, exactly, few | 10.7 |
| Always-Never | always, almost always, sometimes, almost never, never | 10.1 |
| Sizes | larger than, smaller than, same size as | 8.9 |
| Conditional | if, only if | 6.4 |
| Sequence | done in this order, ordered like this | 2.8 |
| Prompt not used | | 14.4 |

Table 1: Relational prompts and their frequencies.

In our game, the initial model is ROBERTA-LARGE [24], fine-tuned on two datasets of binary (yes/no) question, using standard multi-task training: (a) 50K examples from TWENTY QUESTIONS (20Q),[1] a question answering dataset which focuses on commonsense questions such as *"Does an aircraft fly?"* (true) and *"Do everyone have an alarm?"* (false); and 73K automatically-generated yes/no questions from CONCEPTNET triplets, where we transform triplets to assertions with a manually-constructed template for each CONCEPTNET relation. For example, for the triplet (*"wheel"*, *"part-of"*, *"car"*) we generate *"a wheel is part of a car"* (true).

---

[1] https://github.com/allenai/twentyquestions

We re-trained the model five times when the number of (unvalidated) questions collected reached 1,000, 2,000, 5,000, 10,000 and 20,000. The user then gives immediate feedback to the model on whether the model was correct or incorrect, and receives points accordingly ($B2$, $B3$ in Fig. 1).

**Question validation** The last step ensures that composed questions are ones that most people would agree on. Validators answer questions independent of players, and if their answer is different from the answer provided by the question author, then points are deducted from the player, pushing players to ask good questions that can also be answered by other people. This step is also crucial since we train models on collected data and need to make sure examples are not noisy.

In more detail, validators receive a proposed question ($C$ in Fig. 1), and choose a *validation label*. Validation labels indicate if the answer is *"True"* or *"False"*. Other validation labels include *"Don't know"* to indicate they were not able to find an answer, *"Bad question"* for questions that do not make sense, and *"Sensitive"* for questions that involve sensitive topics. Questions are validated multiple times, and a model is trained to determine the final *gold label* based on the validators' selections. We provide more detail on this model in §2.2. To incentivize players to provide truthful answers, we manually label 2,753 questions, and test the users on these questions in 10% of the validations. If players provides an incorrect answer for these questions, they lose 1 point, otherwise they gain 2 points for each validation performed in the game.

**Details of point system** Beating the AI (asking a question the AI gets wrong) grants the players 5 points plus 4 points if the relational prompt was used and an additional 4 points if the topic prompt was used. If the AI is correct, the user receives a default of 3 points for the effort invested. If a question is discarded after quality assurance, 3 points are deducted from the player who authored the question, otherwise, if the final answer is different than the one provided by the player ($B2$), 2 points are deducted. This prevents players from stating that the model is incorrect to earn more points.

## 2.2  Quality Assurance and Dataset Construction

We now describe the details of collecting CSQA2 and the measures taken for quality assurance.

**Automatic question verification** To automatically filter noisy examples, we manually label 2,753 with the labels *"True"*, *"False"*, and *"Bad Question"* (collapsing *"Bad Question"*, *"Don't know"* and *"Sensitive"* above). We then train a linear model to predict the gold label. Features in this linear model include: (a) *Validator features*: The conjunction of validation label, validator accuracy, and validator experience (number of validated questions). For example the feature `Label:True,Acc:High,Exp:Low` indicates that a validator that has high accuracy w.r.t the gold labels, but little experience, indicated that the answer to the assertion is *"True"*; (b) *Player features*: Since players also serve as validators (see below), we featurize players in a similar manner to validators but also includes the prediction of the model-in-the-loop (yes or no).

After training the question validation model, we discard any example that is classified as *"Bad Question"* or questions where the confidence of the verification model is low. Then, we label the question according to the prediction of the question verification model. Out of the questions that pass this verification step, 92% are labeled correctly in the development set of the verification model. Further steps for improving quality assurance are described below.

**Crowdsourcing questions** We used Amazon Mechanical Turk (AMT) workers to compose and validate questions. Workers were designated as *players* 70% of the time, and *validators* the rest of the time. Players receive 4.4$ when they reached 300 points. To increase the quality of the dataset, we only keep questions composed by workers that have a validation accuracy of at least 60% on the expert-labeled examples, and that less than 30% of the questions they author are discarded by the automatic question verification model.

**Adding Google snippets** A possible strategy of malicious players is to find long-tail knowledge facts on the web and ask the model if the fact is correct or not. For example, *"Lobsters taste with their feet"*. To circumvent this, we issue a web query to Google search for every question and take the top-100 returned snippets, as well as the featured Google snippet, if it exists. We then filter out questions for which there is a snippet with a contiguous span that has very low edit-distance to the authored question. This process leads to the filtering of 421 questions from the final dataset. Aside

for quality assurance, in §5 we also investigate whether using Google snippets as additional context can improve the performance of models on CSQA2.

Overall, from the 144,682 questions that were authored, we selected a total of 14,343 examples. The average total cost per question is $0.17 (compared to $0.33 in CommonsenseQA 1.0 [29]).

## 3   Game Analysis

To increase the motivation of players to beat the AI, we experimented with various forms of user feedback. We now analyze how such feedback affects the collected data.

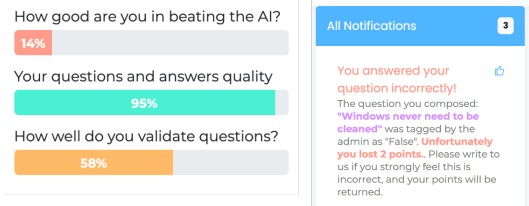

**Feedback on performance**   We experimented with showing players a bar-chart indicating (a) the fraction of authored question where the player beat the AI (AI beat-rate), (b) the fraction of questions that passed the verification model, and (c) the fraction of correct validations w.r.t

Figure 3: Feedback given to users during the game. Left: metrics on average daily player performance. Right: notification on a bad question that leads to point deduction.

the expert-labeled set described above (Fig. 3, left): under 15% in red, under 30% in yellow, and above 30% in green. Interestingly, the average AI beat-rate increased from 26.8% in the 10 days before the introduction of the feedback feature, to 35% in the 10 days immediately after the change.

**Validation feedback**   We provided daily notifications to players, where we showed them questions where validators changed the answer, or questions that were discarded because they were predicted by the automatic question verification model to be a *"Bad Question"* (Fig. 3, right). Moreover, as mentioned in §2.1, we introduced a 3-point penalty for such cases. Notifications led to a 4.8% absolute increase in average positive validations (i.e., the validators agree with the yes/no answer given by the player). However, this also led to questions that are easier for the model – Fig. 4 (dashed black line) shows that after notifications were introduced, the

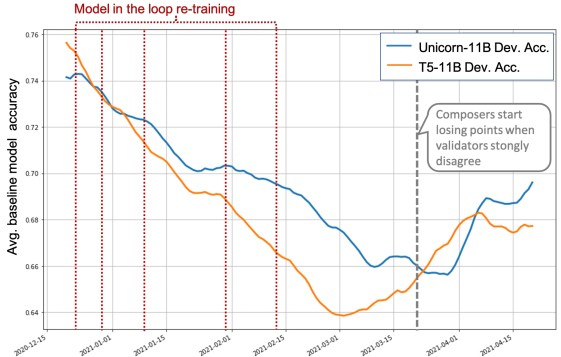

Figure 4: Baselines acc. over question composition time.

average accuracy of models on questions increased, for example, UNICORN-11B [25] improved from 0.66 average accuracy to almost 0.70. This is since players are more hesitant to author questions that may be too difficult for validators. Our analysis emphasizes how subtle design choices in the game can lead to significant ramifications on players' motivation and data quality.

**Model re-training**   We re-trained the model-in-the-loop five times during the data collection period. Fig. 4 shows in red the dates in which the model-in-the-loop was trained, as well as the performance (Y-axis) of our best performing baselines, UNICORN [25] and T5 [26], on development set questions during those dates. Performance drops significantly and consistently from 0.74 to 0.66 as the model-in-the-loop, RoBERTa-Large, is re-trained, indicating that a stronger model drives players to compose more challenging questions, and that the rise in difficulty is shared across pre-trained models.

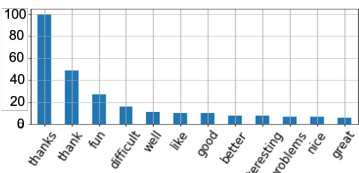

Figure 5: Sentiment words in comments.

**The fun factor**   At the end of each playing session, players were encouraged to leave feedback. We selected sentiment words out of the 100 most frequently used words in the comments, shown in Fig. 5. We find that users enjoy the game and mostly use positive sentiment words. Fun is an important factor in encouraging high engagement, allowing us to select annotators that are better at beating the AI, while maintaining a low average cost.

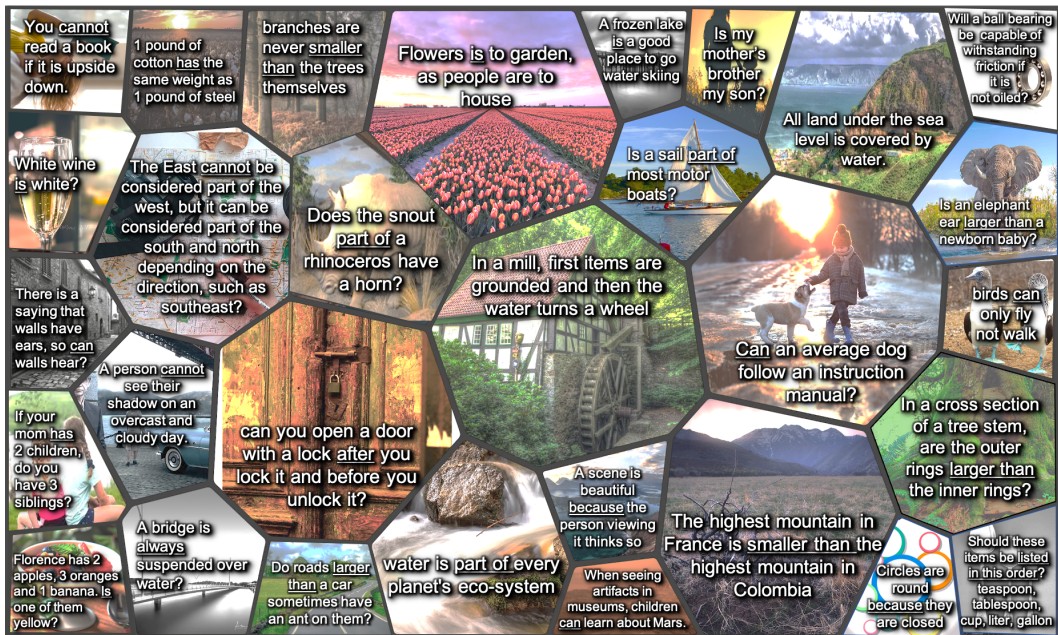

Figure 6: Distribution of the relational prompt words in questions. Each image displays a topic prompt, the area of each image is proportional to the frequency of the corresponding relational prompt in the dataset.

## 4 Dataset Analysis

**Key statistics** Table 2 contains key statistics for CSQA2. The dataset contains $14,343$ questions that are relatively short ($11.3$ words), but include a rich vocabulary ($21,143$ distinct words). The questions are diverse, such that the most common relational prompt appears in only $6.0\%$ of the questions and the most common topic prompt appears in $0.2\%$ of the questions. Overall, CSQA2 includes $1,868$ distinct topic prompts and $33$ distinct relational prompts.

**Question formulation** Overall, $2,537$ AMT players participated in the game, where $351$ were chosen for the final dataset creation, of them $55$ created more than $85\%$ of the questions. For reference, this is 30x more participants compared to COMMONSENSEQA 1.0 [29], indicating the appeal of the game to players. The average annotator session lasted $18$ minutes, composing on average $32.3$ questions, of which $12.2$ the model did not answer correctly, and $26.2$ validations. The majority of annotators ($85.9\%$) were from the USA, $3.3\%$ from India and $2.8\%$ from the UK. A few hundred questions were tagged as *"Sensitive"* by the validating players, because they referred to a topic that may be considered sensitive, such as race, gender or politics. These questions were filtered out.

| Measurement | Value |
|---|---|
| # Distinct Questions | 14,343 |
| % "no" answer | 50.7 |
| # Distinct words in questions | 21,143 |
| Avg. question length (words) | 11.3 |
| Std question length (words) | 6.2 |
| # Participating players | 2,537 |
| # Dataset annotators | 351 |
| Avg. # validations per example | 2.3 |
| # Distinct topic prompt | 1,868 |
| # Distinct relational prompts | 33 |
| % Majority relational prompt | 6.0 |
| % Majority topic prompt | 0.2 |
| % relational prompt used | 85.6 |
| % topic prompt used | 98.3 |

Table 2: Key statistics for CSQA2.

**Topics and relations** We analyzed the distribution of relational prompts and topic prompts in the questions. Fig. 6 visually presents the breakdown, where in each example the relational prompt is underlined, and the sum of the area of all examples with the same relational prompt corresponds to the relative frequency of the relational prompt in the dataset. Only $14.4\%$ of the questions did not use the relational prompt, preferring to beat the AI without gaining the extra point for using it. The most frequently used relational prompts were *"can"* ($5.9\%$ of examples), *"is"* ($5.8\%$), *"cannot"* ($5.7\%$), *"part of"* ($4.9\%$) and *"has"* ($4.1\%$). Only $2\%$ of the questions did not contain the suggested topic prompt and the usage of topics was relatively uniform, with the most frequently used topic, *"January"*, used in only $0.2\%$ of the questions.

| Skill | Description (Example) | % |
|---|---|---|
| Capable of | Whether an object is capable of performing an action (*"A watch is capable of telling the past time"*) | 24.5 |
| Long-tail knowledge | The question contains factual long-tail information (*"Washington DC is located further south than Washington State"*) | 24.5 |
| Plausibility | Quantifiers or always-never relations (*"The peak of a mountain almost always reaches above the the tree line"*) | 23.6 |
| Comparison | Comparison between two objects (*"The end of a baseball bat is larger than the handle"*) | 16.4 |
| Physical | Physical commonsense (*"Do you build the walls on a house before putting on the roof?"*) | 13.6 |
| Causality | Cause and effect relations (*"If you get into an accident because you have been drinking alcohol you will be arrested?"*) | 13.6 |
| Temporal | Temporal understanding (*"None had ever reached the top of Mount Everest before 1977?"*) | 10.0 |
| Negation | The question includes a negation phrase (*"A mock trial is something with no legal consequence"*) | 9.1 |
| Strategy | Reasoning steps are implicit and should be inferred using a strategy (*"Blood banks almost never take cash or checks as deposits"*) | 9.1 |
| Event chain | Question is about order of events (*"Putting on shoes is done in this order normally: person ties shoelaces then slips shoes onto feet"*) | 3.6 |

Table 3: Skills and their frequency in the analyzed data (each example can be annotated with multiple skills).

**Reasoning skills** To analyze the types of reasoning skills needed to correctly answer questions in CSQA2, we randomly sampled 110 examples from the development set and performed the following analysis. For each question, we manually annotated the types of commonsense skills a human would potentially use to correctly answer the question. We annotate multiple skills per question, such that the average question includes 1.48 skills. When annotating the skills, we focus on phenomena that are frequent in our questions, such as plausibility, causality, and negation. Table 3 presents the skill categories used, their frequency in the analyzed examples, a description, and an example.

## 5 Experimental Evaluation

**Experimental setup** We split the data into a training/development/test set containing $9,282/2,544/2,517$ examples, respectively. We preform a *topic prompt split* where the training-set is disjoint from the development and test sets in terms of topic prompts.

**Human evaluation** To test human accuracy, we created a batch of questions for validation (as explained in §2.1), and asked players to validate questions they did not previously see without knowing the feedback given by the player nor the model prediction. Humans obtained an average accuracy of $90.3\%$ per question, and when using the majority vote per question reached an accuracy of $94.1\%$ when comparing to the final answers of the questions (Table 4).

To empirically evaluate that CSQA2 is indeed challenging for state-of-the-art models, we experiment with multiple baselines, as described below.

### 5.1 Baselines

**T5** [26] is a text-to-text model built on top of the transformer architecture [30] pretrained using masked language modeling [31] on a large collection of NLP benchmarks. We conduct experiments with two model sizes, namely 770 million parameters (T5-LARGE) and 11 billion parameters (T5-11B), fine-tuned on the training set of CSQA2.

**UNICORN** [25] is a pretrained commonsense reasoning model obtained by taking T5 and multi-task training on RAINBOW, a suite of commonsense benchmarks. RAINBOW combines commonsense reasoning about social situations and physical interactions, as well as reasoning about the most plausible explanation or continuation

|  | Dev | Test |
|---|---|---|
| GPT-3 | 58.4 | 52.9 |
| T5-LARGE | 53.8 | 54.6 |
| UNICORN-LARGE | 56.4 | 54.9 |
| T5-11B | 68.5 | 67.8 |
| UNICORN-11B | **69.9** | **70.2** |
| Human | 94.1 | - |

Table 4: Development and test accuracies (%) on CSQA2.

of everyday narratives. Pre-trained on the RAINBOW benchmark, UNICORN offers an efficient commonsense reasoning model ready to be fine-tuned on other downstream commonsense tasks. It was shown to reach state-of-the-art performance on 9 different commonsense datasets, including COMMONSENSEQA 1.0. Similar to T5, we report results for UNICORN-LARGE and UNICORN-11B, with 770 million and 11 billion parameters, respectively, fine-tuning models on the training set.

**GPT-3** [1] is a 175B-parameter autoregressive language model trained on a large corpus of web text. While we fine-tune the previous baselines on the full CSQA2 training set, we evaluate GPT-3 in a few-shot inference setting. Namely, we provide a prompt that contains example questions with their respective answers, followed by a question of interest. We then ask GPT-3 to generate

a text completion that corresponds to the answer of the question. In §5.2, we report experiments where the prompt contains $K = 5$ *randomly-selected* (from the training set) example questions. We experimented with multiple values of $K$, as well as a variety of other heuristics for choosing the few-shot examples to be included in the prompt. We provide full details of the prompt format, and the results of these experiments in the supplementary material.

## 5.2 Experimental Results

Table 4 shows results of all models on the development and test sets of CSQA2. T5-LARGE and T5-11B achieve a test accuracy of $54.6\%$ and $67.8\%$, respectively. UNICORN-LARGE and UNICORN-11B obtain a test accuracy of $54.9\%$ and $70.2\%$, respectively. This illustrates how increasing model size leads to dramatic improvements in commonsense capabilities, and that fine-tuning on RAINBOW leads to a sizable increase for the 11B models – from $67.8\%$ to $70.2\%$. GPT-3 achieves a test accuracy of $52.9\%$, well below the T5 and UNICORN baselines, despite being much larger (but without fine-tuning). For all models, performance is significantly lower than human performance, which is at $94.1\%$, leaving ample room for future model improvements.

**Augmenting questions with Google snippets** In the results above, knowledge needed for answering questions must be encoded in the model weights. We now test whether adding Google search results as additional context improves the performance of our T5 and UNICORN baselines. Specifically, we issue a Google web query for each question, and add the top-$k$ snippets as additional context before the question. Table 5 shows the results of this experiment. We see that snippets from Google search queries consistently improve the quality of the predictions of the T5 and UNICORN models, from 67.8 to 72.5 for T5-11B and from 70.2 to 73.3 for UNICORN-11B. This shows that some knowledge on the web is useful for answering questions from CSQA2.

|  | Dev | Test |
|---|---|---|
| T5-11B ($k = 1$) | 73.5 | **73.9** |
| UNICORN-11B ($k = 1$) | 73.9 | 73.4 |
| T5-11B ($k = 5$) | **74.1** | 72.5 |
| UNICORN-11B ($k = 5$) | 74.0 | 73.3 |

Table 5: Development and test accuracies (%) on CSQA2, when including Google search snippets in the input, where $k$ indicates the number of snippets prepended before the question.

## 6 Model Analysis

**Baseline analysis** To analyze baselines on different skills, we examine their performance on different sets of relational prompts that correspond to similar skills (Table 6). For example the *"has"*, *"have"*, *"part of"* and *"is a"* relational prompts correspond to meronymy/hypernymy, *"because"* and *"causes"* correspond to causality, and *"smaller than"*, *"larger than"* and *"same size as"*, correspond to size comparison. We only consider cases where the player used the prompt when composing the question. For skills that cannot be inferred from the relational prompt, such as whether the question contains factual long-tail knowledge, we manually annotate questions in which the skill exists.

We find that T5 and UNICORN do well on questions that include meronymy or hypernymy (Table 6). T5-11B and UNICORN-11B outperform GPT-3 by 19.4 and 18.0 points without Google snippets, and 24.4 and 25.0 with 5 Google snippets, respectively. Accuracy for questions that require causality are higher by 6 points for UNICORN-11B (79.3) compared to T5-11B (73.3), outperforming GPT-3 (49.1) by more than 30 points. Accuracy for examples that require size comparison and long-tail factual knowledge is lower for T5 and UNICORN, such that accuracy for UNICORN-11B is 63.1 and 60.8, respectively. Adding Google snippets increases performance on questions that require long-tail factoid knowledge by as much as 22.5 points.

| Category | Examples | % | T5-11B | Uni-11B | T5-11B ($k = 5$) | Uni–11B ($k = 5$) | GPT-3 |
|---|---|---|---|---|---|---|---|
| Meronymy/ Hypernymy | Q: Scales are an important part of music and fish (A: *yes*) Q: The Earth is a planet that is made primarily of air and helium? (A: *no*) | 14.2 | **75.0** | **73.6** | 80.0 | 80.6 | 55.6 |
| Causality | Q: Saltwater is a chemistry solution because it combines water and salt (A: *yes*) Q: Drinking milk causes bones to become weaker (A: *no*) | 4.6 | **73.3** | **79.3** | 82.8 | 85.3 | **49.1** |
| Long-tail factoid knowledge | Q: Independence is a place in California and Utah (A: *yes*) Q: A male seahorse cannot give birth (A: *no*) | 24.5 | 61.8 | **60.8** | 77.5 | **83.3** | 56.9 |
| Size comparison | Q: The outmost layer of an onion is larger than the inner layers are? (A: *yes*) Q: Electrons are smaller than mesons (A: *no*) | 9.2 | **60.1** | **63.1** | 64.4 | 65.2 | 59.2 |

Table 6: Qualitative error analysis. For each category, we provide two examples and the accuracy.

**Contrast sets**   To examine the consistency of our baselines, we created contrast sets [8, 9], that is, we took 60 questions from the development set and authored additional questions that are minor perturbations of the original question. For example, for the original question *"A bird has 3 wings."*, we add *"A bird has wings"* and *"A bird has only one wing"*, testing if the model understand the exact number of wings birds have.

|  | *Avg.* | *EM* |
|---|---|---|
| Unicorn-11B | **68.8** | **18.3** |
| T5-11B | **68.8** | 11.7 |
| GPT-3 | 53.6 | 3.3 |

Table 7: Diagnostic-set acc.

Overall, we authored 276 questions (4.6 additional examples per original question). Table 7 presents model performance on the contrast set with two evaluation metrics: the average score on all questions (including the original ones), and a *consistency* metric, which checks if a model answers correctly all the questions in each contrast set. All models perform poorly in the consistency setup, Unicorn-11B achieving a score of 18.3, outperforming T5-11B (11.7). GPT-3 scores a close-to-zero result of 3.3.

**GPT-3 analysis**   To better understand GPT-3's behaviour, we replace our yes/no prompts with questions that have free-form answers. We then manually label whether GPT-3's free form answers (FF) reflect understanding of the question and answer (see examples in Table 8). We find that in 10% of the cases GPT-3 answers *"Unknown"*. Of the remaining questions, only 61% of the answers reflect agreement with the original model trained with yes/no prompts. Of the questions where GPT-3's answer agrees with its original prediction, we add the word *"why"* as a prefix to the question and find that only 36.4% of the explanations make sense. This hints to the lack of consistency in the representation of knowledge of GPT-3.

| Q: Question.  FF: GPT-3 free-form prediction. Why: GPT-3 generative prediction when adding the word *"why"* as a prefix to the question. |
|---|
| Q:*"The office is capable of making people laugh?"*  FF:*"Yes"*✔ Why:*"The office is capable of making people laugh because it is a comedy."*✔ |
| Q:*"Some people don't seem to mind not breathing for an hour?"*  FF:*"They are called free divers."*✘ Why:*"Because they are dead."*✘ |
| Q:*"When water freezes, does it get hard?"*  FF:*"No, water expands when it freezes."*✘ Why:*"Water expands as it freezes, so it doesn't get hard."*✘ |

Table 8: Analysis of GPT-3 predictions.

**Google snippets analysis**   We analyzed GPT-3 performance on questions whose string was found in a Google web snippet, but for which the answer is *"no"*. Interestingly, we find that in many of these cases GPT-3 does not distinguish between imaginary or counterfactual quotes and real-life facts, answering *"yes"* to questions such as *"Is it true that all the leaves are brown and the sky is gray?"*, *"Is it true that shadows have shadows?"* and *"Is it true that You've Got to Hide Your Love Away?"*. This alludes to the challenges involved in separating factual from imaginary content on the web.

## 7   Related Work

Our work joins a line of previously introduced benchmarks targeting commonsense reasoning [12, 13, 29, 15] as well as probing for other reasoning abilities [32, 33, 34, 35, 36, 37, 38]. An emerging trend in NLP dataset creation is the use of a model-in-the-loop when humans compose examples: a model is used either as a post-hoc filter or during annotation. Examples for this include [39, 40, 41, 12, 42, 34, 43, 38, 44], with recent works featuring re-training during dataset collection (Beat-the-AI [22], StrategyQA [23] and Dynabench [21]). This approach makes the resulting dataset more challenging to current models, driving research to tackle new problems. Gamification, the collection of data through a fun interactive design, was attempted by prior work for multi-choice question answering [45], image labeling [46] and protein folding [47]. Contemporarily, Fool Me Twice [28] introduced a game for collecting entailment data through multi-player interaction. Our work focuses on gamifying the interaction between a human and a model.

## 8   Conclusion

In this work, we propose gamification as a general framework for creating diverse and challenging NLU benchmarks. We use this framework to collect CSQA2, a new benchmark that contains 14,343 yes/no questions. We perform a detailed analysis of CSQA2, which elucidates the unique properties of our dataset, and thoroughly evaluate on a strong suite of baselines. We find that the best model, Unicorn-11b, achieves an accuracy of 70.2%, dozens of points lower than human accuracy. We argue that gamification is a promising approach for creating challenge sets that expose the weaknesses of current state-of-the-art models.

# 9 Acknowledgments

We thank our colleagues at The Allen Institute of AI. This research was partially supported by The Yandex Initiative for Machine Learning and the European Union's Seventh Framework Programme (FP7) under grant agreement no. 802800-DELPHI.

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
