# OpenReview forum: "CommonsenseQA 2.0: Exposing the Limits of AI through Gamification"
_NeurIPS.cc/2021/Track/Datasets_and_Benchmarks/Round1 — NeurIPS 2021 Datasets and Benchmarks Track (Round 1)_

### Official Review · Reviewer_xbJB · 2021-07-03
**Testing AI systems with human-generated commonsense questions  via a gamification approach**

**Rating:** 10
**Confidence:** 3

**Strengths:**

I liked the proposed game-based framework, the design is simple and seems effective, and the validation framework is neat.
It looks like a promising approach to generating scalable commonsense-knowledge datasets for testing AI systems.

The evaluation of 'baseline' SOTA systems is impressive, and results are very interesting. It quite amazing that GPT-3 performs so poorly on this dataset.

**Weaknesses:**

I don't see any essential weaknesses in this paper.

**Additional Feedback:**

A few typos:
line 33: "we proposes"
line 112: weaknss."
line 141: "players provides"


Questions to authors:
1.
How did you deal with typos (and grammar, etc) in players' questions?
2.
Have you encountered cases of almost identical questions from different players?

**Clarity:**

The paper is clear and well written. It is also easy to understand as the description is high-level, without complex technical details. This it should be easy for a wide audience.

**Correctness:**

This is a rather novel aproach to constructing a common-sense dataset - as a gae rather than an annotation task. The use of the neat verification subtask seems to be crucial to obtaining good data.

**Documentation:**

A large part of the paper is devoted to describing how this dataset was collected.
The supplementary material contains the full documentation and hosting details.
Reproducibility of the baseline results is supported as the authors provide details of parameters for the systems.

**Ethics:**

I find no ethical problems with this paper. The authors also describe a procedure for eliminating sensitive questions in the dataset generation process.

**Relation To Prior Work:**

The paper is based on prior work by same authors, and also relevant related work is presented and cited.

**Summary And Contributions:**

The paper describes an approach to involing humans in  generating challenging (to AI) questions, a framework that is scalable, provides rich sets of various questions and can be partially stirred by developers. The approach amounts to a game where human participans need to generate yes/no questions to beat an AI system (player gets more points if system provides wrong answer to their question). Final insentive for palyers was monetary reward (when reaching a certain amount of points). An additional important aspect of this framework is validation of questions - questions are validated by other players  - which is designed to ensure that players do not 'game' the system.
The AI system in the game was based on ROBERTA language model.
The game allowed collecting a large number of validated questions (14K).

The paper then describes how various SOTA systems perform on this dataset: the best system achieves 70% correct, whereas humans achieve about 94% correct. The reported results can be taken as baselines for future reserach with this dataset.

The authors also present various analyses of the dataset and insight into the game-based framework that enabled its creation.

This is a very creative work and the paper provides nice analyses of the results and systems.

---

### Official Review · Reviewer_SGXE · 2021-07-03
**This paper creates a new QA dataset (CSQA2), which is more challenging and diverse. The analysis of the proposed dataset is comprehensive and promising as well.**

**Rating:** 7
**Confidence:** 4

**Strengths:**

1. The authors propose a new QA dataset, containing more challenging and diverse data compared with previous datasets.
2. To create collect data, they design a new gamification framework by using the model-in-the-loop approach, which ensures the diversity and difficulty of the collected data as well.
3. Besides, to analyse the constructed dataset, the authors consider several widely used natural language understanding (NLU) methods (i.e., GPT-3, T5, and UNICORN).


**Weaknesses:**

1. In Section 2.2, each validator feature contains three labels: a validation label, a validation accuracy, and a validation experience. Here, the validation label refers to the answer of the question (i.e., True or False) while the validation accuracy indicates whether the validator has high accuracy w.r.t. the gold label. But I am a little confused about the meaning of validation experience, and I am not sure how it works during the question verification processing.
2. In Table 5, after incorporating the top-k snippets from Google, the existing methods (T5 and UNICORN) achieve a significant improvement. However, why do the quantitative results change slightly when increasing $k=1$ to $k=5$?
3. A more detailed comparison between the proposed QA dataset and existing ones seems necessary, such as the number of questions, or the average length of questions. Summarise in a table may be a good choice.


**Additional Feedback:**

N/A

**Clarity:**

The paper is well written and organised, especially for the descriptions of the data collection part, which makes it easy to follow.

**Correctness:**

Most of the claims in this paper are correct. The processing of dataset construction seems reasonable and technical sound. Specifically, the processing of data collection can be divided into three parts: controlled question generation, model-in-the-loop, question validation, which ensure both the variety and quality of the collected data. To filter out the unsatisfied data, the authors adopt three strategies, i.e., automatic question verification, crowdsourcing questions verification, and adding google snippets. Note that the method of google snippets seeks to remove the questions that contain long-tail knowledge.

**Documentation:**

The paper provides relatively sufficient details on both data collection and data organisation. Moreover, the authors provide a link, which contains the dataset and code.

**Ethics:**

To avoid the problem of ethics, the authors design a “sensitive” item when formulating each question, which distinguishes whether the validated question would cause ethics problems, such as race, gender, or politics.

**Relation To Prior Work:**

This paper well discusses the differences between the proposed CSQA2 dataset and its previous version [29]. Besides, the authors also provide more discussions about the similarity and differences for data collection and processing in Section 7.

**Summary And Contributions:**

This paper collects a challenging and diverse QA dataset, called COMMONSENSEQA 2.0 (CSQA2), containing 14,343 yes/no questions. For data collection, the authors propose a new gamification framework inspired by a model-in-the-loop approach. Based on the constructed dataset, they conduct a detailed analysis to investigate the unique properties of the dataset. Besides, they test a series of widely used QA baselines (e.g., GPT-3, T5, and UNICORN) on the proposed dataset, and expose the weaknesses of the existing methods.

---

### Official Review · Reviewer_SJrS · 2021-07-05
**Rigorous treatment to different aspects involved in gamification and to ensure quality; good paper, accept**

**Rating:** 7
**Confidence:** 3
**Correctness:** All details and claims seem correct t…
**Clarity:** Very well written

**Strengths:**

1. Very well written
2. Sounds rigorous and complete in its treatment to possible malicious acts by the players. The authors have been careful to address many issues that could appear in such a data creation exercise
3. Carefully thought experiments to establish the utility of the dataset in challenging the state-of-the-art models.
4. Good model analysis to understand the strengths and weaknesses of existing models

**Weaknesses:**

1. Missing details on and comparison with CommonSenseQA 1.0
2. Would have loved to see some ablation studies to verify the impact of certain elements of the gamification which have been introduced for specific purposes. For example, impact of using Google snippets on the quality of data collected; impact of topic prompt or relational prompt in the questions; impact of using different models for model-in-the-loop as compared to the model bein evaluated.


**Additional Feedback:**

Nothing beyond what is mentioned above.

**Documentation:**

Documentation seems sufficient (not in the main paper, but on the web)

**Ethics:**

There doesn't seem to be any. The authors have taken care of modeling the sensitivity of a question.

**Relation To Prior Work:**

This is one of the areas which need improvement. It also lacks some details on and comparison with CQA 1.0

**Summary And Contributions:**

1. The authors have created a challenging and diverse QA dataset containing 14343 examples
2. They have proposed a new gamification based framework for data creation and to ensure quality
3. They demonstrate that the state-of-the-art models fall short as compared to humans, when evaluated on their challenging dataset

---

### Decision · Program_Chairs · 2021-07-26

**Decision:**

Accept

**Comment:**

All the reviewers agree that the paper proposes an interesting benchmark along with a gamification and model-in-the loop based framework for question answering. This framework seems to be effective for building more creative and challenging benchmarks. Therefore, I recommend acceptance of this paper.